# Muscle magnetic resonance characterization of STIM1 tubular aggregate myopathy using unsupervised learning

Amalia Lupi[1]ᵒ*, Simone Spolaor[2]ᵒ, Alessandro Favero[1], Luca Bello[3], Roberto Stramare[4], Elena Pegoraro[3]‡, Marco Salvatore Nobile[5]‡*

**1** Institute of Radiology, Department of Medicine–DIMED, University of Padua, Padua, Italy, **2** Microsystems, Department of Mechanical Engineering, Eindhoven University of Technology, Eindhoven, The Netherlands, **3** Department of Neurosciences, University of Padua, Padua, Italy, **4** Clinical and Translational Advanced Imaging Unit, Department of Medicine–DIMED, University of Padua, Padua, Italy, **5** Department of Environmental Sciences, Informatics and Statistics (DAIS), Ca' Foscari University of Venice, Venice, Italy

ᵒ These authors contributed equally to this work.
‡ EP and MSN also contributed equally to this work.
* amalia.lupi@phd.unipd.it (AL); marco.nobile@unive.it (MSN)

**Data Availability Statement:** All relevant data are within the manuscript and its Supporting Information files.

## Abstract

### Purpose

Congenital myopathies are a heterogeneous group of diseases affecting the skeletal muscles and characterized by high clinical, genetic, and histological variability. Magnetic Resonance (MR) is a valuable tool for the assessment of involved muscles (i.e., fatty replacement and oedema) and disease progression. Machine Learning is becoming increasingly applied for diagnostic purposes, but to our knowledge, Self-Organizing Maps (SOMs) have never been used for the identification of the patterns in these diseases. The aim of this study is to evaluate if SOMs may discriminate between muscles with fatty replacement (S), oedema (E) or neither (N).

### Methods

MR studies of a family affected by tubular aggregates myopathy (TAM) with the histologically proven autosomal dominant mutation of the STIM1 gene, were examined: for each patient, in two MR assessments (i.e., t0 and t1, the latter after 5 years), fifty-three muscles were evaluated for muscular fatty replacement on the T1w images, and for oedema on the STIR images, for reference. Sixty radiomic features were collected from each muscle at t0 and t1 MR assessment using 3DSlicer software, in order to obtain data from images. A SOM was created to analyze all datasets using three clusters (i.e., 0, 1 and 2) and results were compared with radiological evaluation.

### Results

Six patients with TAM STIM1-mutation were included. At t0 MR assessments, all patients showed widespread fatty replacement that intensifies at t1, while oedema mainly affected the muscles of the legs and appears stable at follow-up. All muscles with oedema showed

**Funding:** The authors received no specific funding for this work.

**Competing interests:** The authors have declared that no competing interests exist.

fatty replacement, too. At t0 SOM grid clustering shows almost all N muscles in Cluster 0 and most of the E muscles in Cluster 1; at t1 almost all E muscles appear in Cluster 1.

## Conclusion

Our unsupervised learning model appears to be able to recognize muscles altered by the presence of edema and fatty replacement.

## Introduction

Congenital myopathies are a group of genetically determined diseases affecting skeletal muscle and characterized by a spectrum of typical pathological changes within the myofiber.

Clinically, these diseases present with various degrees of muscle weakness, hypotonia, hyporeflexia, poor muscle bulk and sometimes dysmorphic features like pectus carinatum, scoliosis, foot deformities or facial abnormalities.

Because of the high overlap with other muscle conditions, clinical characteristics, family history [1], electromyographic study, muscle biopsy (with histological, immunohistochemical and electron microscopy examinations), muscle magnetic resonance imaging (MRI) and some laboratory tests, are necessary to reach a diagnosis [2].

Tubular Aggregates Myopathy (TAM) due to *STIM1* (stromal interaction molecule 1) gene mutations is a congenital myopathy, clinically characterized by skeletal muscle weakness, myalgias and cramps. Mutations in *STIM1* result in a constitutive channel activation leading to transmembrane calcium flow impairment and the formation of tubular aggregates in skeletal muscle [3].

Muscle biopsies in TAM are characterized by the presence of "tubular aggregates", cellular inclusions consisting of highly ordered arrays of tubules derived from the sarcoplasmic reticulum [4].

STIM1-TAM follows an autosomal dominant pattern of inheritance. STIM1 mutations have also been observed in Stormoken syndrome characterized by bleeding tendency, hematological abnormalities, asplenia, ichthyosis, and congenital miosis [5].

Muscle imaging is emerging as an important tool in the differential diagnosis of congenital myopathies due to its high soft tissue contrast and its ability to detect skeletal muscle abnormalities in trophism and muscle signal.

MRI protocols should include T2w fat suppressed (or STIR, often preferred for less prominent artifacts) sequences to allow radiologists to detect any abnormal increase in muscle signal, corresponding to muscle oedema. Muscle oedema could be caused by trauma, denervation, or myositis, but in the correct clinical setting indicates the acute phase of muscle involvement typical of certain myopathies.

T1w sequences are acquired to detect an increase in muscle signal representing fibro-adipose degeneration, the chronic stage of myopathies; these sequences also allow the detection of morphologic changes, such as the loss of muscle bulk or muscle enlargement (adaptive hypertrophy or pseudohypertrophy related to fibro-adipose replacement) [6], and the type of intramuscular disorganization of muscle texture [7].

Specific patterns are most recognizable in patients presenting mild phenotypes, with individual muscles selectively affected. Extensive and severe muscle involvement, as well as very mild and initial involvement do not allow for clear pattern detection even if they can be identified by expert radiologist [8].

Muscle MRI to be helpful in the diagnostic process needs to be properly interpreted and the radiologist needs to gain experience to correctly evaluate the muscle alterations severity and characteristic imaging pattern.

Radiomics, converting images in data, could be useful in standardizing these assessments. Radiomics predicts that the information reflecting pathophysiologic changes in tissues and organs contained in medical images can be revealed through quantitative analyses.

The basic image features of intensity, shape, size, volume, and texture can be processed via multiple steps, including image acquisition, volume of interest identification, segmentation, extraction and quantification will form a dataset, and mining these datasets, in combination or not with additional demographic, clinical or genomic information will help the diagnostic process.

The principal application of radiomics has so far been used in oncology, but recently it has been applied to other clinical settings.

We employed Self-Organizing Maps (SOMs), an unsupervised machine learning method based on neural networks, [9] to characterize radiomic data obtained from the MRI assessment of a homogeneous, STIM1-TAM cohort of patients, in order to verify the possibility to differentiate healthy muscle (N), fibro-adipose substituted muscle (S), and oedematous muscle (E).

## Material and methods

The study was approved by our Institutional Ethics Committee (316n/AO/22).

### Dataset creation

Seven patients segregating the autosomal dominant p.Leu98Val STIM1 mutation from a three generations family affected by TAM were studied.

Anamnestic data were collected, and medical records reviewed. All data and images were fully anonymized before analysis.

Patients who underwent at least two MR evaluation (i.e., at baseline and follow-up) were included. All of them provide an informed written consent. Upper right girdle and lower limbs MRI scans were all performed with 1.5 T commercial scanner (Magnetom Avanto, Siemens, Erlangen, Germany) at the Institute of Radiology of Padua; T1w and STIR sequences of scapular and forearm girdle of the right hemisoma (for technical convenience) and of the lower limbs were acquired on the axial plane with an 8 mm slice thickness.

A radiologist (AL) reviewed the images of each exam, as reference for subsequent SOM results interpretation. Muscles included in the field of view were evaluated assessing fibro-adipose replacement in T1w sequences using the Mercuri 6 point scale (i.e., stage 0, normal appearance; stage 1, mildly moth-eaten appearance with occasional scattered T1 hyperintense areas; stage 2A, moderate moth-eaten appearance with numerous scattered T1 hyperintense areas; stage 2B, severely moth-eaten appearance with numerous confluent areas of T1 hyperintensity; stage 3, confluent areas of signal hyperintensity and stage 4, complete fatty degeneration with replacement of muscle by connective tissue and fat) [10], and muscle oedema in STIR sequences, using a 5 point scale (0, absent; 1, mild, interfascicular; 2A, mild, intrafascicular, segmented; 2B, mild, intrafascicular, global; 3A, moderate, intrafascicular, segmented; 3B, moderate, intrafascicular, global), already used at our Radiology Institute [11] (Fig 1).

Texture information were obtained by the same radiologist using 3DSlicer open source software (version 4.10.2) by selecting a round-shaped ROI of a standard dimension (1% of screen dimension in total matrix) in the T1w sequences and a round-shaped ROI of the same dimension in the STIR sequences for each muscle assessed at baseline MR (i.e., t0) and at follow-up MR (i.e., t1), at the middle third of the muscle belly, excluding perimysium, tendons and, on

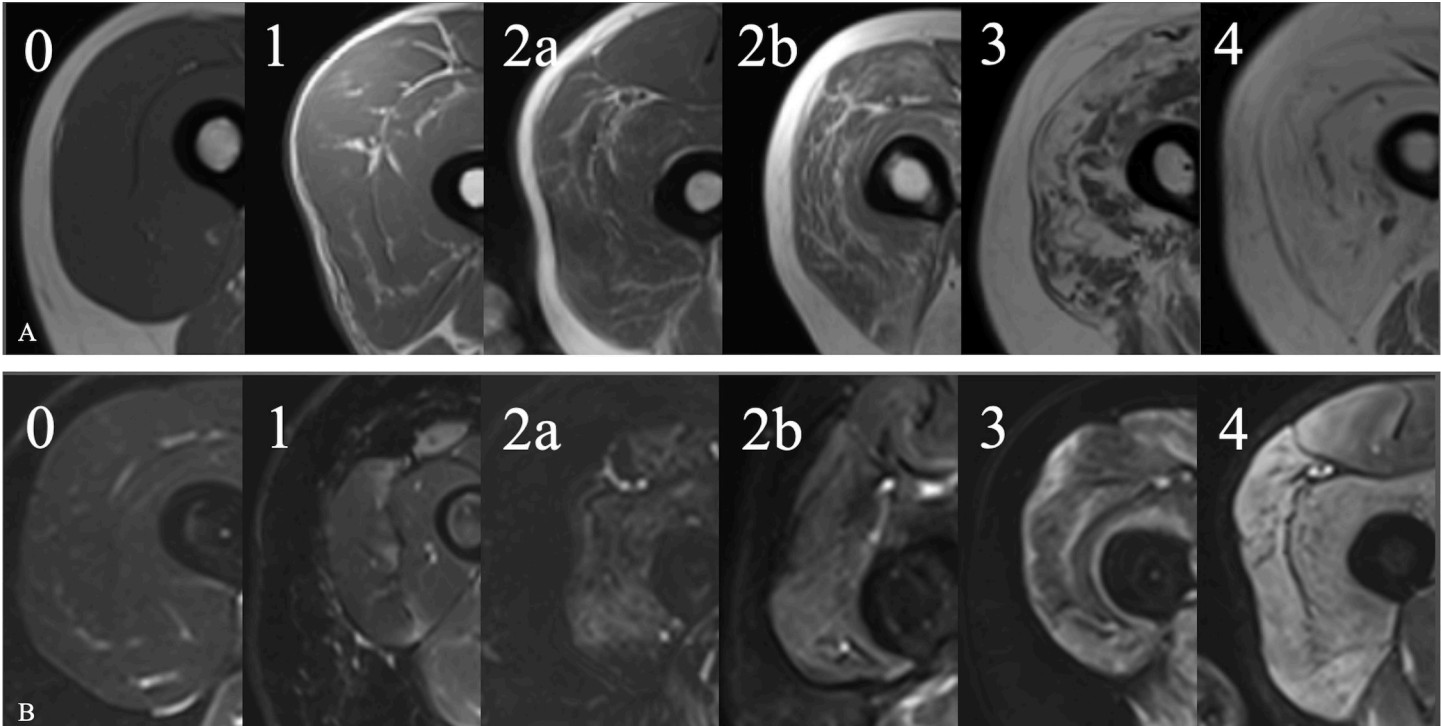

**Fig 1. Representative images of T1- and T2-weighted images classification used.** (A) 6 point scale for fatty infiltration evaluation. (B) 5 point scale for oedema assessment.

STIR images, prominent vascular structures. Features extracted through each ROI were: first order statistic (i.e., describing distribution of values of singular voxels without spatial relationship), Gray Level Run Length Matrix (GLRLM), and Gray Level Co-occurrence Matrix (GLCM). In a preliminary phase of data pre-processing, every included patient was assigned a number, and non-standard data (i.e., Mercuri stages 2a, 2b) were converted in consecutive numbers to obtain machine-learning "friendly" data. Subsequently, for each muscle we created two datasets for MR images, one at t0 and one at t1. The datasets include all radiomics features extracted from T1w and STIR sequences. The two datasets can be found in S1 File.

## Self-organizing map and clustering

Both datasets were processed with an unsupervised neural network. Specifically, we exploited a Self-Organizing Map (SOM, also known as Kohonen's map) [9]. The SOM is a special class of fully connected neural network able to represent high-dimensional data into a low-dimensional (generally 2D) map, in which each cell is associated with a neuron and its set of weights. So doing, at the end of the unsupervised learning process, neurons that are topologically close fire in response to samples (i.e., muscles) sharing similar characteristics. Once trained, the weights of the neurons can be clustered by means of a clustering algorithm, in order to group together samples sharing similar characteristics.

We used this approach to group muscles into three clusters, sharing similar radiomics features. Specifically, we trained a 10x10 SOM with hexagonal topology, employing a gaussian neighborhood function with $\sigma = 1.0$, learning rate $t = 0.5$, and cosine distance. The weights of the SOM were initialized by means of a principal component analysis (PCA), and the training was performed for a total of 500 iterations. The SOM was implemented using Python (version

3.7.4) and the MiniSom library (version 2.2.7) [12]. To cluster the SOM's weights, we employed an agglomerative clustering algorithm with euclidean distance and Ward linkage criterion, implemented with the Scikit-learn Python library (version 1.0.2) [13]. The script employed to generate the results of this work can be found in S1 File. The resulting clustering was compared with each radiological assessment, to verify the SOM reliability in recognizing muscles altered by the presence of oedema and fatty replacement.

## Results

### Clinical and radiological analysis

One patient among the seven family members affected by STIM1-TAM, was excluded from analyses because missed the follow up MR scan. The remaining patients were three females and three males: a woman (73 years-old at baseline MR evaluation-patient II-3), her 2 sons (52 and 50 years-old at baseline MR evaluation, patients III-3 and III-4, respectively), her two daughters (47 and 54 years-old at baseline MR evaluation, patients III-1 and III-6, respectively), and her grandson (21 years-old at baseline MR evaluation, patient IV-3) (Fig 2).

All six patients underwent two MR evaluations: at baseline (t0) and after 5 years (t1) between 2009 and 2019.

Radiological evaluation required about 25 minutes for each examination. 53 muscles for each patient were classified.

In all patients, a pattern of diffuse muscular involvement, with predominant fibro-adipose changes and mild oedema was revealed. Thighs and legs muscles were invariably more affected than arms, with diffuse fibro-adipose replacement of anterior and posterior thighs muscles, particularly of glutei, tensors fasciae latae, quadriceps, adductors, semimembranous and semitendinosus and long heads of biceps femoris.

There was relative sparing of the short head of the biceps femoris in four patients and of gracilis in two patients.

In lower legs a diffuse distribution of fibro-fatty infiltration was noticed, with a predominant involvement of the posterior compartment muscles, mainly soleus and gastrocnemius, flexor hallucis longus and the peroneal muscles; tibialis anterior, extensors digitorum and extensor hallucis were less involved (Fig 3A).

Upper girdles scans revealed mild fibro-adipose changes in deltoid, triceps brachii and biceps brachii only in the three more severely affected patients (III-1, III-3, III-4) while subscapularis muscle was involved in all patients with different degrees of severity (Fig 3B).

From t0 to t1 the general pattern of muscle involvement did not differ, but in patients III-3 and III-6 there was a worsening of MRI fibro-fatty replacement that was concurrent to a clinical worsening of muscle strength. Fibro-fatty replacement and clinical worsening were observed also in patient III-4 and, more prominently, in patient II-3. Patient III-1 did not show any progression at muscle MRI but a worsening of muscle strength clinical evaluation, where patient IV-3 muscle strength was unchanged at follow-up but an increase in fibro-adipose replacement was observed.

Muscle oedema assessment revealed at t0 minimal positive findings in the thighs and lower legs in III-1, mild and moderate involvement respectively in the thighs and lower legs in patients IV-3 and III-3, minimal involvement in lower legs in patient II-3 and in upper girdle muscles in III-6. No oedema was observed in patient III-4; at t1 patients IV-3 and III-3 underwent a slight worsening in the same anatomical regions involved at t0, patient III-4 became positive with appearance of minimal edematous findings in the thighs and lower legs, and patients II-3, III-1 and III-6 were stable.

All muscles with oedema also showed fibro-adipose replacement.

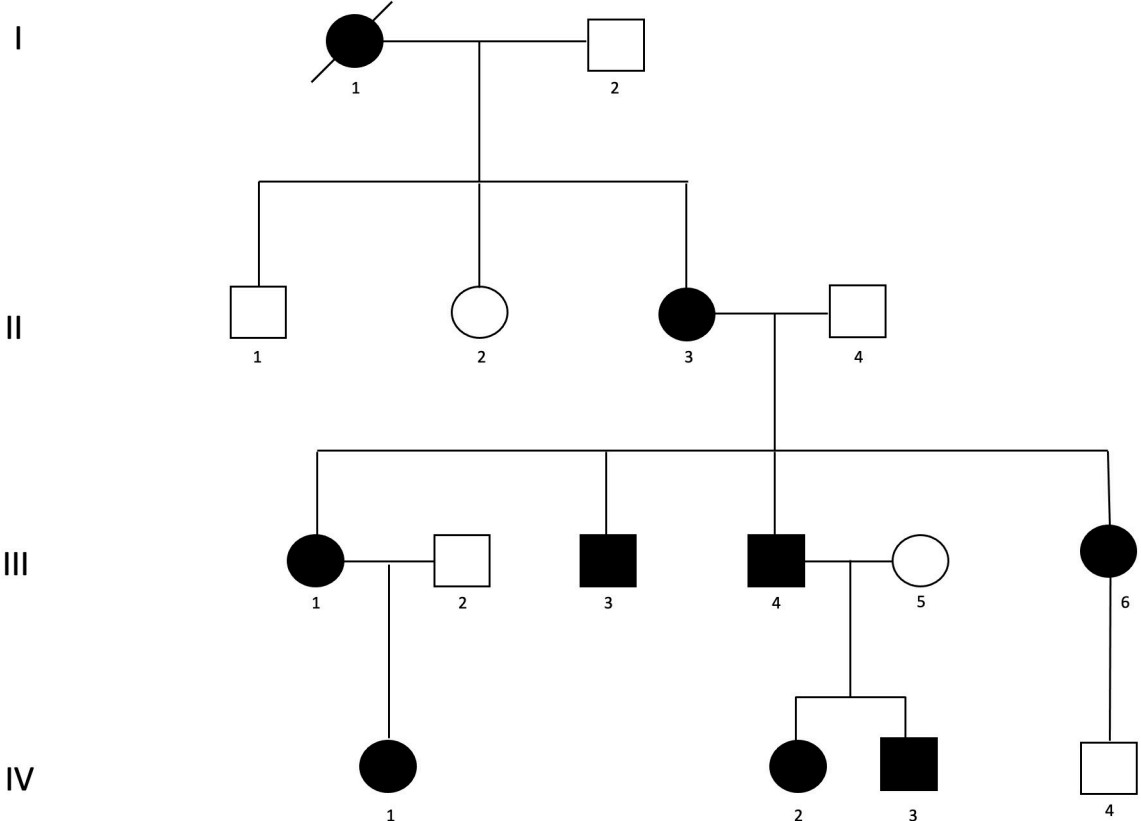

**Fig 2. STIM1-TAM family pedigree.** Black squares, affected males; black circles, affected females; crossed out black circle, affected female deceased; white squares and circles, unaffected males and females, respectively.

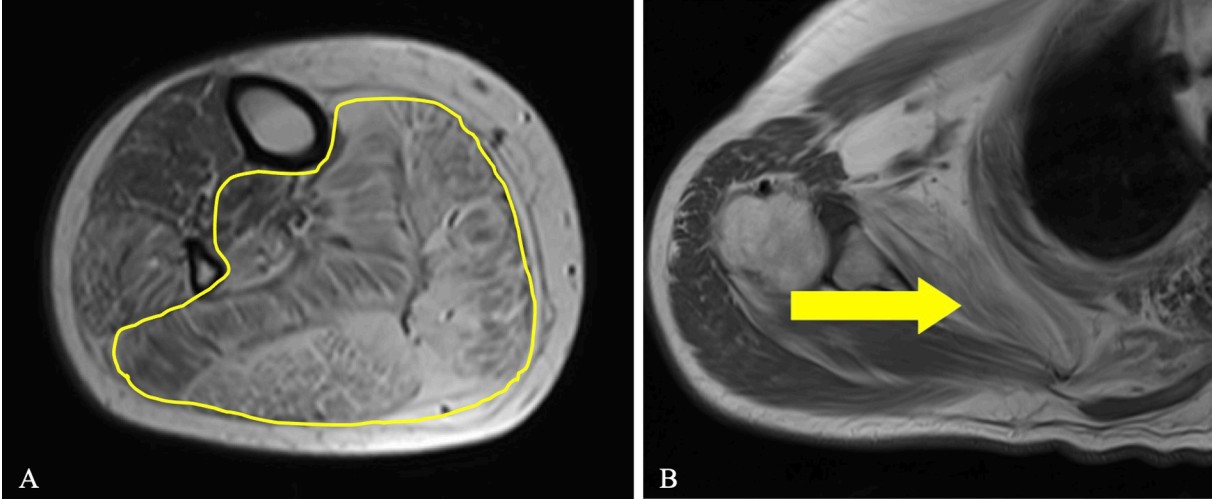

**Fig 3. Fibro-fatty infiltration pattern representative images.** (A) Axial T1-weighted MR image of the leg showing muscle fibro-fatty infiltration with predominant involvement of posterior muscles (yellow line), such as soleus, medial and lateral gastrocnemii, and flexor hallucis longus. (B) T1-weighted MR image of the shoulder girdle showing the involvement of subscapularis muscle (yellow arrow).

### Radiomics and artificial intelligence analysis

Due to excessive artifact presence (mainly movement or peripheral magnetic field inhomogeneities artifacts), it was considered appropriate to exclude 22 muscles from t0 evaluation and 10 from the t1 evaluation for the extraction of radiomics data. For each muscle, at t0 and t1 MR evaluation, fifty-eight features (i.e., 18 first order statistics, 16 GLRLM, and 24 GLCM) were extracted from T1w and STIR images. The procedure required about 30 minutes for each examination.

A separate SOM was trained for each dataset, and used to partition the data into three clusters, according to the radiologically determined muscles categories: normal muscle (N, Mercuri and oedema scale 0 points), fibro-adipose replacement (S, Mercuri scale 1–4 points), fibro-adipose replacement and oedema (E, oedema score 1-3b and Mercuri scale 1–4 points). The results obtained on the t0 dataset are shown in Fig 4.

SOM output was compared to human radiological evaluation. At t0, 74% of normal muscles (N) and 68% of muscles with fatty replacement and oedema (E) were clustered together; at t1, 97% of "E" muscles were in two out of three clusters (Fig 5). Since in our SOMs the muscles displaying fibro-adipose substitution (S) were distributed in all clusters we decided to perform a further analysis, using only two clusters (Fig 6). Excluding the samples classified as S, this new SOM was able to correctly classify 91.4% of the oedema samples. A full overview of the classification performance on E and N muscles is available in Table 1. Normal muscles were correctly classified in 50% of cases. It should be noted that they were underrepresented in our dataset.

## Discussion

Patterns of muscle involvement is becoming critical in the diagnostic process of muscle diseases. We report the muscle MRI pattern in 6 patients affected by STIM1-mutated TAM and compare radiologist evaluation with the output of a clustering based on Self-Organizing Maps. The typical pattern of STIM1-TAM includes in the lower limbs an overall fat replacement of the anterior and posterior thighs, calf, and peroneal muscles, with almost invariably reported involvement of flexor hallucis longus muscle and sparing of tibialis anterior; other features often reported were sartorius muscle involvement with sparing of gracilis and short head of biceps femoris [14–17]. In the upper limbs the subscapularis muscle and often the lumbar extensors are involved with sparing of trapezius and masticatory muscles.

Despite the different location of *STIM1* gene mutations, and the small number of patients reported this pattern of muscle involvement is common indicating in muscle MRI a valuable tool for diagnosis of this rare myopathy [16].

In our cohort of STIM1-TAM patients, flexor hallucis longus and subscapularis muscles were almost always affected, confirming the known pattern of muscle involvement and the usefulness of evaluating specific muscles that rarely are involved in the pathological process [14], in the differential diagnosis with many others muscle diseases (i.e., alpha dystroglycanopathies, calpainopathy, *TPM2*-related and *RYR1*-related myopathies, among the others).

Moreover, the previously reported sparing of gracilis and tibialis anterior were not constantly observed in our series, even if they were never the most affected muscles [14–17].

The findings of progression of muscle fibro-fatty infiltration and the subtle variations of muscle oedema from t0 to t1 MR scans confirmed that STIM1-TAM is a progressive muscle disease [5]. In general muscle MR involvement parallels muscle strength, being the more MR involved muscles the weakest, as shown in patients III-3, III-4, III-6 and II-3; on the contrary, when fatty infiltration or muscle weakness are mild, a direct correlation between the two is difficult to observe: patient IV-3 MRI assessment showed an increase in fibro-fatty replacement

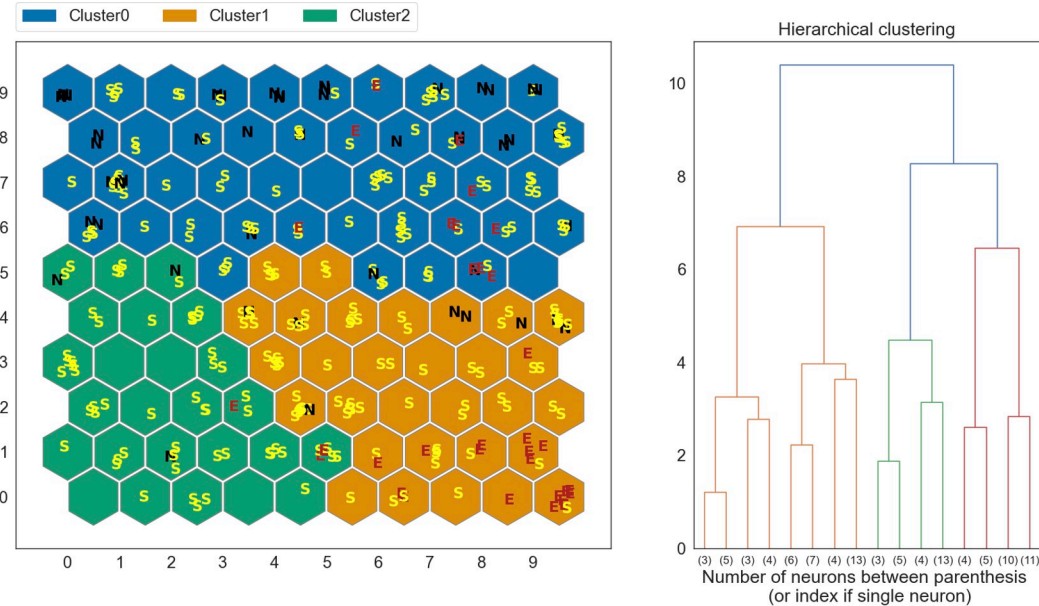

**Fig 4. Clustering results obtained on a SOM trained on the t0 dataset.** The clusters are compared to labels (E, S, N) assigned by trained human evaluators.

at t1 in hamstring, flexor hallucis longus, and flexor digitorum muscles that was not consistent with the muscle strength stability, and patient III-1 showed a progression of muscle weakness but a stability of fibro-adipose replacement at MRI.

Given the emerging importance of muscle MR in the diagnostic process of congenital myopathies, the development of a computed automatic single muscle scoring system is becoming relevant. In the literature multiple attempts of applying texture analyses on specific

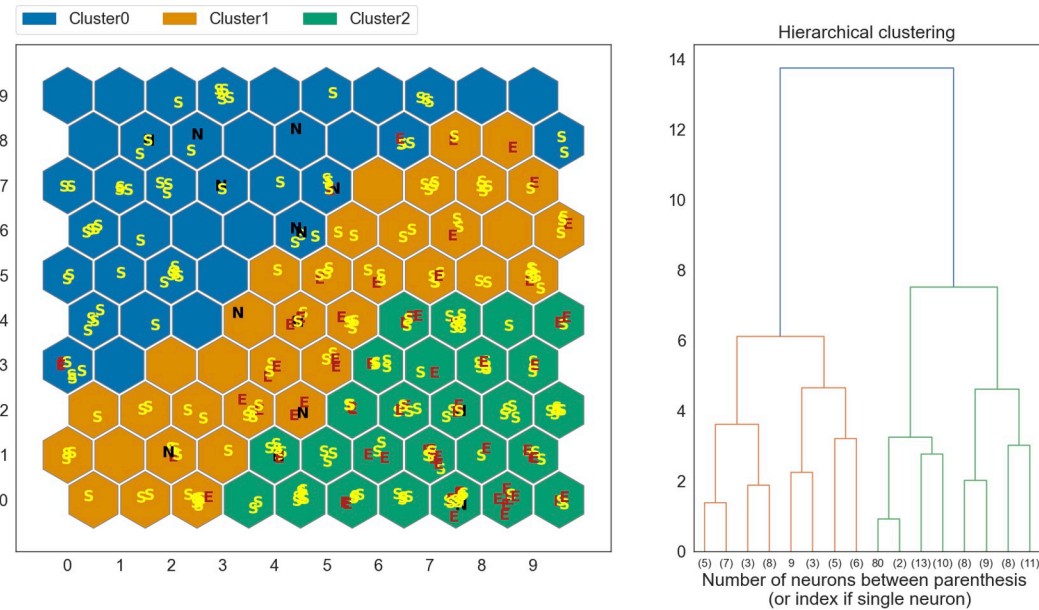

**Fig 5. Clustering results obtained on a SOM trained on the t1 dataset.** The clusters are compared to labels (E, S, N) assigned by trained human evaluators.

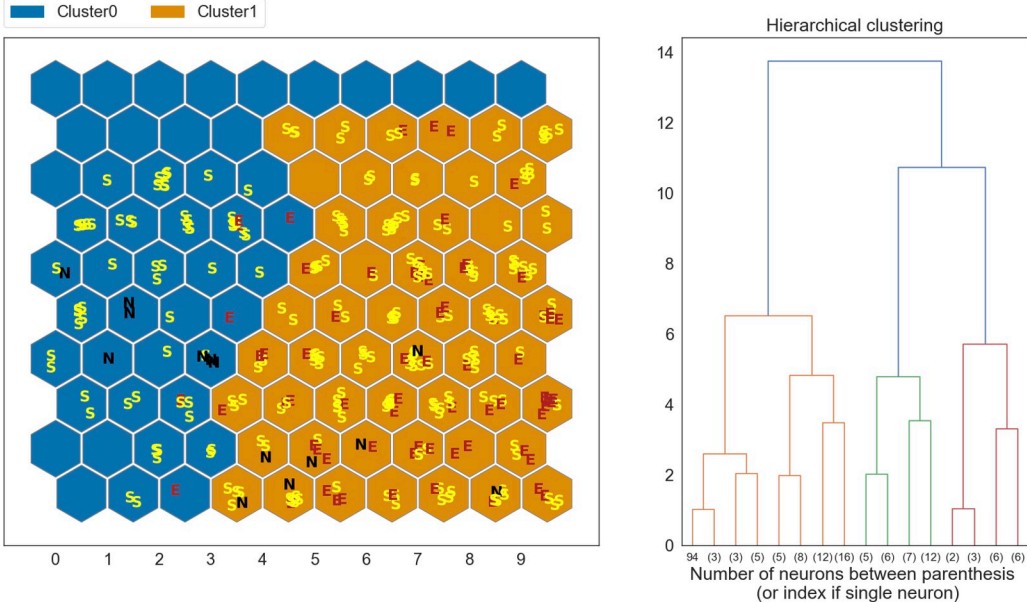

**Fig 6. Clustering results obtained by retraining a SOM on the t1 dataset using only 2 clusters.** The clusters are compared to labels (E, S, N) assigned by trained human evaluators.

myopathy are reported and showed, for example, their capability in distinguishing normal and pathologic muscles in neurogenic and myopathic diseases [18], in distinguishing inflammatory myopathy from myotonic dystrophy [19], identifying the progression of disease [20], quantifying muscle T2 water and fat fraction in facio-scapulo-humeral dystrophy [21], and in discerning between different types of inflammatory myopathies [22].

The main goal of our study was to evaluate SOMs performance in distinguishing radiological muscles appearance in order to identify disease progression or stability in our STIM1-TAM cohort.

At t0 SOM attributed most of healthy (N), and fibro-adipose and oedema (E) muscles to two different clusters, underlining that a trained SOM can recognize muscles with pathologic MRI alterations.

At t1 evaluation, a more evenly distribution of the healthy and involved muscles among the three clusters, could suggest that, with general disease progression, a SOM might recognize alterations also in muscles classified as normal by human's eye and human evaluation might overestimate negative samples from MRI image inspection; indeed, the SOM was still able to discern "E" muscles and to separate them in the additional analysis.

SOMs are a non-supervised machine learning method that allowed us to subdivide muscles in clusters without a human expert evaluation. SOMs can capture the non-linear correlations often existing in biological and medical datasets, and their output can be more easily

**Table 1. Classification performance over the N and E samples, using two clusters on the t1 dataset.**

|  | Predicted normal (N) | Predicted oedema (E) |
|---|---|---|
| True normal (N) | 7 (50.0%) | 7 (50.0%) |
| True oedema (E) | 6 (8.6%) | 64 (91.4%) |

Percentage over total amount of samples belonging to that class reported between parentheses.

interpreted by professionals not belonging to the computer science field, thanks to the fact that they visualize their outcomes through "maps".

Although with our dataset the SOM did not discriminate perfectly between our three radiological categories, it was able to discriminate between normal and affected muscles.

Further analyses with larger cohorts are needed to confirm our results and to carry out other applications, such as in pattern recognition of congenital myopathies.

A limit of this study is represented by the selection of a ROI for radiomic data extraction; the dimension of the ROI was chosen to be the same for all muscles. This allows the analyses of multiple muscles in a feasible time, being the amount of time necessary for the process of image segmentation one of the main issues in Radiomics texture analyses [23].

## Conclusions

We argue that a trained SOM can recognize muscles altered by the presence of oedema and fatty replacement.

However, in our experience, human expertise can better evaluate also different aspects, such as artifacts impact on images.

Further analyses will serve to confirm whether a SOM can detect alterations invisible to the human eye.

## Supporting information

**S1 File.**
(ZIP)

## Author Contributions

**Conceptualization:** Amalia Lupi, Simone Spolaor, Marco Salvatore Nobile.

**Data curation:** Amalia Lupi, Simone Spolaor, Alessandro Favero, Elena Pegoraro.

**Formal analysis:** Simone Spolaor.

**Investigation:** Amalia Lupi, Alessandro Favero, Luca Bello, Roberto Stramare, Elena Pegoraro.

**Methodology:** Amalia Lupi, Simone Spolaor, Marco Salvatore Nobile.

**Resources:** Amalia Lupi, Simone Spolaor, Marco Salvatore Nobile.

**Software:** Amalia Lupi, Simone Spolaor.

**Supervision:** Elena Pegoraro, Marco Salvatore Nobile.

**Validation:** Luca Bello, Roberto Stramare, Elena Pegoraro, Marco Salvatore Nobile.

**Writing – original draft:** Amalia Lupi, Simone Spolaor, Alessandro Favero.

**Writing – review & editing:** Amalia Lupi, Elena Pegoraro, Marco Salvatore Nobile.

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
