## [Decision Letter · Decision Letter 0]

21 Feb 2023

PONE-D-22-30569Muscle magnetic resonance characterization of STIM1 tubular aggregate myopathy using unsupervised learningPLOS ONE

Dear Dr. Lupi,

Thank you for submitting your manuscript to PLOS ONE. After careful consideration, we feel that it has merit but does not fully meet PLOS ONE’s publication criteria as it currently stands. Therefore, we invite you to submit a revised version of the manuscript that addresses the points raised during the review process.

We look forward to receiving your revised manuscript.

Kind regards,

Atsushi Asakura, Ph.D

Academic Editor

PLOS ONE

Journal Requirements:

3. Please provide the specific name of the ethics committee/IRB that approved your study, and the approval number.

Reviewers' comments:

Reviewer's Responses to Questions

**Comments to the Author**

1. Is the manuscript technically sound, and do the data support the conclusions?

Reviewer #1: Yes

Reviewer #2: Partly

2. Has the statistical analysis been performed appropriately and rigorously? 

Reviewer #1: Yes

Reviewer #2: I Don't Know

3. Have the authors made all data underlying the findings in their manuscript fully available?

Reviewer #1: Yes

Reviewer #2: No

4. Is the manuscript presented in an intelligible fashion and written in standard English?

Reviewer #1: Yes

Reviewer #2: Yes

5. Review Comments to the Author

Reviewer #1: Dear authors,

The manuscript you submitted aims to characterize the defect observed in STIM1 myopathy by MR.

The unsupervised method (SOM) is correct, and the data presented solid. Unfortunately, the data does not seem to correlate with the clinical severity observed in patients. I am favorable to propose the acceptance of your paper once the minor comments are considered.

Minor comment:

In the abstract you propose to classify the dystrophic parameter in F and O, whereas in the text the proposed classification is S and E?

Line 206 : Please correct parallealed by parallel.

Line 240 : Figure 2 missed some information to be knowledgeable. In fact, all figure legends need to be more precise with more information.

Reviewer #2: Tubular aggregate myopathy (TAM) is a progressive muscle disorder pathologically proven with the presence of tubular aggregates in affected muscle tissues. Lupi A et al. studied the change in MR findings of 53 muscles of six patients with TAM twice. The authors assessed and divided the findings into three categories: fatty replacement (the 6‐grade Mercuri scale), oedema (a five-point scale), or neither. The obtained data were processed to make Self-Organizing Map (SOM) of three or two clusters that can discriminate between normal and affected muscles. Although the discrimination ability is not so poor, the reviewer wonders how to apply this SOM in a clinical setting. Clinicians expect that this method can tell which muscle is affected. The presented data are not sufficient to show the usefulness of the developed machine learning method.

In spite of the description: All relevant data are within the manuscript and its Supporting Information files; the reviewer can not find multidimensional source numerical data and supporting information.

The data that support the statements “flexor hallucis longus and subscapularis muscles were almost always affected” in lines 268-269, and “sparing of gracilis and tibialis anterior were not constantly observed in our series” in lines 273-274 are not shown. Please prepare the data table/list of the magnitude of change (such as fatty replacement and oedema) in each muscle.

The description of “human evaluation might overestimate negative samples from MRI image inspection” in lines 236-237 seems too speculative. Please move this statement to Discussion section. The trained artificial intelligence will not work properly if the original dataset is improper. The reviewer believes that the prediction will not be superior to the original data

The evaluation of prediction ability should be conducted using distinctive (test) datasets, otherwise, the authors can not show to what extent the method can predict in general.

Minor points

The term ‘SOMs’ first appears in line 41. Please spell it out as Self-Organizing Maps in this line but not in line 116.

Neither the Mercuri scale nor the Five-point scale of muscle oedema in STIR sequences is noted in Fig. 2 and 3. In addition, the schematic illustrations or images with leader lines will be helpful for readers not familiar with MR findings.

There are some misspellings ‘withsparing’ in line 264 and ‘contraywise’ in line 278

6. PLOS authors have the option to publish the peer review history of their article (what does this mean?). If published, this will include your full peer review and any attached files.

Reviewer #1: No

Reviewer #2: No

---

## [Author Response · Author response to Decision Letter 0]

5 Apr 2023

Dear Editor, 

We thank you for your consideration of our manuscript “Muscle magnetic resonance characterization of STIM1 tubular aggregate myopathy using unsupervised learning” (PONE-D-22-30569). 

We would like to thank the Reviewers for their suggestions, which allowed us to improve our manuscript. In particular, we extended the description of the figures and their captions, as well as providing more information about the data on which our unsupervised learning method was trained upon. We also made the train data publicly available.

We prepared a point-by-point response to all the comments raised by the Reviewers. All changes implemented in the manuscript have been highlighted in red color. 

We hope that, with these modifications, you will find our manuscript suitable for publication in the PLOS ONE journal. 

Yours sincerely, 

Amalia Lupi, on behalf of all co-authors 

 

Reviewer #1

The manuscript you submitted aims to characterize the defect observed in STIM1 myopathy by MR. The unsupervised method (SOM) is correct, and the data presented solid. Unfortunately, the data does not seem to correlate with the clinical severity observed in patients. I am favorable to propose the acceptance of your paper once the minor comments are considered.

Authors’ response: We thank the reviewer for their valuable comments.

In the abstract you propose to classify the dystrophic parameter in F and O, whereas in the text the proposed classification is S and E?

Authors’ response: Thanks for your kind notification. We apologize for this incorrect description. The misclassification in the abstract has been corrected

Line 206 : Please correct parallealed by parallel.

Authors’ response: We rephrased the sentence indicated by the reviewer, in order to improve its readability.

Line 240 : Figure 2 missed some information to be knowledgeable. In fact, all figure legends need to be more precise with more information.

Authors’ response: We modified Figure 2 (now Figure 3a) and extended its caption to improve its understandability. Moreover, we extended the captions of Figure 4, 5 and 6.

Reviewer #2

Tubular aggregate myopathy (TAM) is a progressive muscle disorder pathologically proven with the presence of tubular aggregates in affected muscle tissues. Lupi A et al. studied the change in MR findings of 53 muscles of six patients with TAM twice. The authors assessed and divided the findings into three categories: fatty replacement (the 6‐grade Mercuri scale), oedema (a five-point scale), or neither. The obtained data were processed to make Self-Organizing Map (SOM) of three or two clusters that can discriminate between normal and affected muscles. Although the discrimination ability is not so poor, the reviewer wonders how to apply this SOM in a clinical setting. Clinicians expect that this method can tell which muscle is affected. The presented data are not sufficient to show the usefulness of the developed machine learning method.

Authors’ response: We thank the reviewer for their valuable comments.

In spite of the description: All relevant data are within the manuscript and its Supporting Information files; the reviewer can not find multidimensional source numerical data and supporting information.

Authors’ response: Thank you for this comment. We provided the full dataset and the code used for generating the results in the supplementary files of the manuscript.

The data that support the statements “flexor hallucis longus and subscapularis muscles were almost always affected” in lines 268-269, and “sparing of gracilis and tibialis anterior were not constantly observed in our series” in lines 273-274 are not shown. Please prepare the data table/list of the magnitude of change (such as fatty replacement and oedema) in each muscle.

Authors’ response: Thank you for this comment. We provided the full dataset. 

The description of “human evaluation might overestimate negative samples from MRI image inspection” in lines 236-237 seems too speculative. Please move this statement to Discussion section.

Authors’ response: We thank the reviewer for this suggestion.

The trained artificial intelligence will not work properly if the original dataset is improper. The reviewer believes that the prediction will not be superior to the original data.

Authors’ response: We provided the training dataset and the code as supplementary files. Moreover, we better clarify the evaluation process in the reply to the following comment of this letter.

The evaluation of prediction ability should be conducted using distinctive (test) datasets, otherwise, the authors can not show to what extent the method can predict in general.

Authors’ response: We would like to point out that the method present in our paper is unsupervised, and it does not use any information from the labels and clinicians classification during the training process. This method was chosen and made necessary because of the restricted size of the available dataset (comprising 6 patients, with data collected for 53 muscles each). To test the effectiveness of our unsupervised method, we compared its outcome (i.e. clustering of muscles) with the classification provided by clinicians on MR images. In this way, we can evaluate the performance of our method with external ground truth.

The term ‘SOMs’ first appears in line 41. Please spell it out as Self-Organizing Maps in this line but not in line 116.

Authors’ response: We spelled out the acronym as suggested by the reviewer.

Neither the Mercuri scale nor the Five-point scale of muscle oedema in STIR sequences is noted in Fig. 2 and 3. In addition, the schematic illustrations or images with leader lines will be helpful for readers not familiar with MR findings.

Authors’ response: Thanks for your kind suggestion. We have modified and implemented Figures 2 and 3 (now Figure 3 a and b), also introducing representative images of the applied classifications (now Figure 1).

There are some misspellings ‘withsparing’ in line 264 and ‘contraywise’ in line 278

Authors’ response: We corrected the indicated misspelled words.

---

## [Editor Report · Decision Letter 1]

24 Apr 2023

Muscle magnetic resonance characterization of STIM1 tubular aggregate myopathy using unsupervised learning

PONE-D-22-30569R1

Dear Dr. Lupi,

We’re pleased to inform you that your manuscript has been judged scientifically suitable for publication and will be formally accepted for publication once it meets all outstanding technical requirements.

Kind regards,

Atsushi Asakura, Ph.D

Academic Editor

PLOS ONE
---

## [Editor Report · Acceptance letter]

27 Apr 2023

PONE-D-22-30569R1 

Muscle magnetic resonance characterization of STIM1 tubular aggregate myopathy using unsupervised learning 

Dear Dr. Lupi:

I'm pleased to inform you that your manuscript has been deemed suitable for publication in PLOS ONE. Congratulations! Your manuscript is now with our production department. 

Kind regards, 

on behalf of

Dr. Atsushi Asakura 

Academic Editor

PLOS ONE